# LEARNING IDENTITY MAPPINGS WITH RESIDUAL GATES

**Pedro H. P. Savarese**
COPPE/PESC
Federal University of Rio de Janeiro
Rio de Janeiro, Brazil
savarese@land.ufrj.br

**Leonardo O. Mazza**
Poli
Federal University of Rio de Janeiro
Rio de Janeiro, Brazil
leonardomazza@poli.ufrj.br

**Daniel R. Figueiredo**
COPPE/PESC
Federal University of Rio de Janeiro
Rio de Janeiro, Brazil
daniel@land.ufrj.br

## ABSTRACT

We propose a layer augmentation technique that adds shortcut connections with a linear gating mechanism, and can be applied to almost any network model. By using a scalar parameter to control each gate, we provide a way to learn identity mappings by optimizing only one parameter. We build upon the motivation behind Highway Neural Networks and Residual Networks, where a layer is reformulated in order to make learning identity mappings less problematic to the optimizer. The augmentation introduces only one extra parameter per layer, and provides easier optimization by making degeneration into identity mappings simpler. Experimental results show that augmenting layers provides better optimization, increased performance, and more layer independence. We evaluate our method on MNIST using fully-connected networks, showing empirical indications that our augmentation facilitates the optimization of deep models, and that it provides high tolerance to full layer removal: the model retains over 90% of its performance even after half of its layers have been randomly removed. In our experiments, augmented plain networks – which can be interpreted as simplified Highway Neural Networks – perform similarly to ResNets, raising new questions on how shortcut connections should be designed. We also evaluate our model on CIFAR-10 and CIFAR-100 using augmented Wide ResNets, achieving 3.65% and 18.27% test error, respectively.

## 1 INTRODUCTION

As the number of layers of neural networks increase, effectively training its parameters becomes a fundamental problem (Larochelle et al. (2009)). Many obstacles challenge the training of neural networks, including vanishing/exploding gradients (Bengio et al. (1994)), saturating activation functions (Xu et al. (2016)) and poor weight initialization (Glorot & Bengio (2010)). Techniques such as unsupervised pre-training (Bengio et al. (2007)), non-saturating activation functions (Nair & Hinton (2010)) and normalization (Ioffe & Szegedy (2015)) target these issues and enable the training of deeper networks. However, stacking more than a dozen layers still lead to a hard to train model.

Recently, models such as Residual Networks (He et al. (2015b)) and Highway Neural Networks (Srivastava et al. (2015)) permitted the design of networks with hundreds of layers. A key idea of these models is to allow for information to flow more freely through the layers, by using shortcut connections between the layer's input and output. This layer design greatly facilitates training, due to shorter paths between the lower layers and the network's error function. In particular, these models can more easily learn identity mappings in the layers, thus allowing the network to be deeper

and learn more abstract representations (Bengio et al. (2012)). Such networks have been highly successful in many computer vision tasks.

On the theoretical side, it is suggested that depth contributes exponentially more to the representational capacity of networks than width (Eldan & Shamir (2015) Telgarsky (2016) Bianchini & Scarselli (2014) Montúfar et al. (2014)). This agrees with the increasing depth of winning architectures on challenges such as ImageNet (He et al. (2015b) Szegedy et al. (2014)).

Increasing the depth of networks significantly increases its representational capacity and consequently its performance, an observation supported by theory (Eldan & Shamir (2015) Telgarsky (2016) Bianchini & Scarselli (2014) Montúfar et al. (2014)) and practice (He et al. (2015b) Szegedy et al. (2014)). Moreover, He et al. (2015b) showed that, by construction, one can increase a network's depth while preserving its performance. These two observations suggest that it suffices to stack more layers to a network in order to increase its performance. However, this behavior is not observed in practice even with recently proposed models, in part due to the challenge of training ever deeper networks.

In this work we aim to improve the training of deep networks by proposing a layer augmentation that builds on the idea of using shortcut connections, such as in Residual Networks and Highway Neural Networks. The key idea is to facilitate the learning of identity mappings by introducing a shortcut connection with a linear *gating mechanism*, as illustrated in Figure 1. Note that the shortcut connection is controlled by a gate that is parameterized with a scalar, $k$. This is a key difference from Highway Networks, where a tensor is used to regulate the shortcut connection, along with the incoming data. The idea of using a scalar is simple: it is easier to learn $k = 0$ than to learn $W_g = 0$ for a weight tensor $W_g$ controlling the gate. Indeed, this single scalar allows for stronger supervision on lower layers, by making gradients flow more smoothly in the optimization.

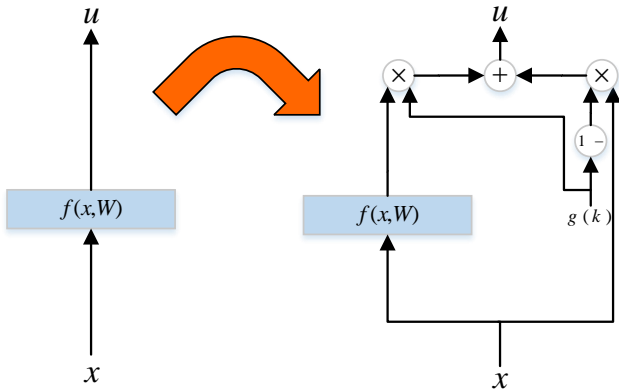

Figure 1: Gating mechanism applied to the shortcut connection of a layer. The key difference with Highway Networks is that only a scalar $k$ is used to regulate the gates instead of a tensor.

We apply our proposed layer re-design to plain and residual layers, with the latter illustrated in Figure 2. Note that when augmenting a residual layer it becomes simply $u = g(k)f_r(x, W) + x$, where $f_r$ denotes the layer's residual function. Thus, the shortcut connection allows the input to flow freely without any interference of $g(k)$ through the layer. In the next sections we will call augmented plain networks (illustrated in Figure 1) Gated Plain Network and augmented residual networks (illustrated in Figure 2) Gated Residual Network, or GResNet. Again, note that in both cases learning identity mappings is much easier in comparison to the original models.

Note that layers that degenerated into identity mappings have no impact in the signal propagating through the network, and thus can be removed without affecting performance. The removal of such layers can be seen as a transposed application of sparse encoding (Glorot et al. (2011)): transposing the sparsity from neurons to layers provides a form to prune them entirely from the network. In-

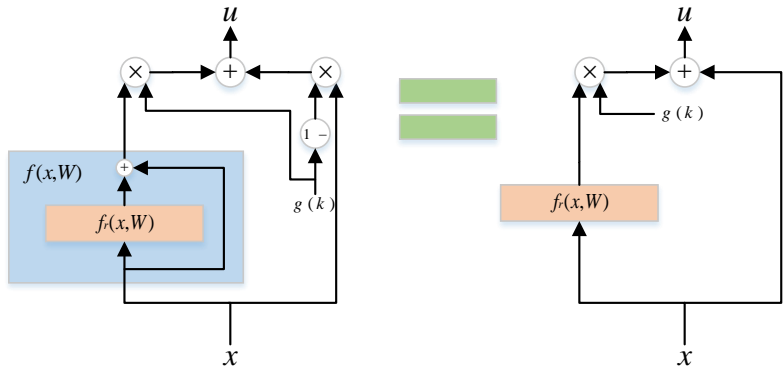

Figure 2: Proposed network design applied to Residual Networks. Note that the joint network design results in a shortcut path where the input remains unchanged. In this case, $g(k)$ can be interpreted as an amplifier or suppressor for the residual $f_r(x, W)$.

deed, we show that performance decays slowly in GResNets when layers are removed, even when compared to ResNets.

We evaluate the performance of the proposed design in two experiments. First, we evaluate fully-connected Gated PlainNets and Gated ResNets on MNIST and compare them with their non-augmented counterparts, showing superior performance and robustness to layer removal. Second, we apply our layer re-design to Wide ResNets (Zagoruyko & Komodakis (2016)) and test its performance on CIFAR, obtaining results that are superior to all previously published results (to the best of our knowledge). These findings indicate that learning identity mappings is a fundamental aspect of learning in deep networks, and designing models where this is easier seems highly effective.

## 2 AUGMENTATION WITH RESIDUAL GATES

### 2.1 THEORETICAL INTUITION

Recall that a network's depth can always be increased without affecting its performance – it suffices to add layers that perform identity mappings. Consider a plain fully-connected ReLU network with layers defined as $u = ReLU(\langle x, W \rangle)$. When adding a new layer, if we initialize $W$ to the identity matrix $I$, we have:

$$u = ReLU(\langle x, I \rangle) = ReLU(x) = x$$

The last step holds since $x$ is an output of a previous ReLU layer, and $ReLU(ReLU(x)) = ReLU(x)$. Thus, adding more layers should only improve performance. However, how can a network with more layers learn to yield performance superior than a network with less layers? A key observation is that if learning identity mapping is easy, then the network with more layers is more likely to yield superior performance, as it can more easily recover the performance of a smaller network through identity mappings.

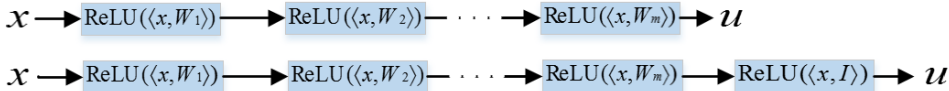

Figure 3: A network can have layers added to it without losing performance. Initially, a network has $m$ ReLU layers with parameters $\{W_1, \dots, W_m\}$. A new, $(m+1)$-th layer is added with $W_{m+1} = I$. This new layer will perform an identity mapping, therefore the two models are equivalent.

The layer design of Highway Neural Networks and Residual Networks allows for deeper models to be trained due to their shortcut connections. Note that in ResNets the identity mapping is learned

when $W = 0$ instead of $W = I$. Similarly, a Highway layer can degenerate into an identity mapping when the gating term $T(x, W_T)$ equals zero for all data points. Since learning identity mappings in Highway Neural Networks strongly depends on the choice of the trasnform function $T$ (and is non-trivial when $T$ is the sigmoid function, since $T^{-1}(0)$ is not defined) we will focus our analysis on ResNets due to their simplicity. Considering a residual layer $u = ReLU(\langle x, W \rangle) + x$, we have:

$$u = ReLU(\langle x, 0 \rangle) + x = ReLU(0) + x = x$$

Intuitively, residual layers can degenerate into identity mappings more effectively since learning an all-zero matrix is easier than learning the identity matrix. To support this argument, consider weight parameters randomly initialized with zero mean. Hence, the point $W = 0$ is located exactly in the center of the probability mass distribution used to initialize the weights.

Recent work (Zhang et al. (2016)) suggests that the L2 norm of a critical point is an important factor regarding how easily the optimizer will reach it. More specifically, residual layers can be interpreted as a translation of the parameter set $W = I$ to $W = 0$, which is more accessible in the optimization process due to its inferior L2 norm.

However, assuming that residual layers can trivially learn the parameter set $W = 0$ implies ignoring the randomness when initializing the weights. We demonstrate this by calculating the expected component-wise distance between $W_o$ and the origin. Here, $W_o$ denotes the weight tensor after initialization and prior to any optimization. Note that the distance between $W_o$ and the origin captures the effort for a network to learn identity mappings:

$$E\left[(W_o - 0)^2\right] = E\left[W_o^2\right] = Var\left[W_o\right]$$

Note that the distance is given by the distribution's variance, and there is no reason to assume it to be negligible. Additionally, the fact that Residual Networks still suffer from optimization issues caused by depth (Huang et al. (2016a)) further supports this claim.

Some initialization schemes propose a variance in the order of $O(\frac{1}{n})$ (Glorot & Bengio (2010), He et al. (2015a)), however this represents the distance for each individual parameter in $W$. For tensors with $O(n^2)$ parameters, the total distance – either absolute or Euclidean – between $W_o$ and the origin will be in the order of $O(n)$.

## 2.2 RESIDUAL GATES

As previously mentioned, the key contribution in this work is the proposal of a layer augmentation technique where learning a single scalar parameter suffices in order for the layer to degenerate into an identity mapping, thus making optimization easier for increased depths. As in Highway Networks, we propose the addition of gated shortcut connections. Our gates, however, are parameterized by a single scalar value, being easier to analyze and learn. For layers augmented with our technique, the effort required to learn identity mappings does not depend on any parameter, such as the layer width, in sharp contrast to prior models.

Our design is as follows: a layer $u = f(x, W)$ becomes $u = g(k)f(x, W) + (1 - g(k))x$, where $k$ is a scalar parameter. This design is illustrated in Figure 1. Note that such layer can quickly degenerate by setting $g(k)$ to 0. Using the ReLU activation function as $g$, it suffices that $k \leq 0$ for $g(k) = 0$.

By adding an extra parameter, the dimensionality of the cost surface also grows by one. This new dimension, however, can be easily understood due to the specific nature of the layer reformulation. The original surface is maintained on the $k = 1$ slice, since the gated model becomes equivalent to the original one. On the $k = 0$ slice we have an identity mapping, and the associated cost for all points in such slice is the same cost associated with the point $\{k = 1, W = I\}$: this follows since both parameter configurations correspond to identity mappings, therefore being equivalent. Lastly, due to the linear nature of $g(k)$ and consequently of the gates, all other slices $k \neq 0, k \neq 1$ will be a linear combination between the slices $k = 0$ and $k = 1$.

In addition to augmenting plain layers, we also apply our technique to residual layers. Although it might sound counterintuitive to add residual gates to a residual layer, we can see in Figure 2 that our augmentation provides ResNets means to regulate the residuals, therefore a linear gating

mechanism might not only allow deeper models, but could also improve performance. Having the original design of a residual layer as:

$$u = f(x, W) = f_r(x, W) + x$$

where $f_r(x, W)$ is the layer's residual function – in our case, **BN-ReLU-Conv-BN-ReLU-Conv**. Our approach changes this layer by adding a linear gate, yielding:

$$
\begin{aligned}
u &= g(k)f(x, W) + (1 - g(k))x \\
&= g(k)(f_r(x, W) + x) + (1 - g(k))x \\
&= g(k)f_r(x, W) + x
\end{aligned}
$$

The resulting layer maintains the shortcut connection unaltered, which according to He et al. (2016) is a desired property when designing residual blocks. As $(1 - g(k))$ vanishes from the formulation, $g(k)$ stops acting as a dual gating mechanism and can be interpreted as a flow regulator. Note that this model introduces a single scalar parameter per layer block. This new dimension can be interpreted as discussed above, except that the slice $k = 0$ is equivalent to $\{k = 1, W = 0\}$, since an identity mapping is learned when $W = 0$ in ResNets.

## 3 EXPERIMENTS

All models were implemented on Keras (Chollet (2015)) or on Torch (Collobert et al. (2011)), and were executed on a Geforce GTX 1070. Larger models or more complex datasets, such as the ImageNet (Russakovsky et al. (2015)), were not explored due to hardware limitations.

### 3.1 MNIST

The MNIST dataset (Lecun et al. (1998)) is composed of $60,000$ greyscale images with $28 \times 28$ pixels. Images represent handwritten digits, resulting in a total of 10 classes. We trained four types of fully-connected models: classical plain networks, ResNets, Gated Plain networks and Gated ResNets.

The networks consist of a linear layer with 50 neurons, followed by $d$ layers with 50 neurons each, and lastly a softmax layer for classification. Only the $d$ middle layers differ between the four architectures – the first linear layer and the softmax layer are the same in all experiments.

For plain networks, each layer performs dot product, followed by Batch Normalization and a ReLU activation function.

Initial tests with pre-activations (He et al. (2016)) resulted in poor performance on the validation set, therefore we opted for the traditional **Dot-BN-ReLU** layer when designing Residual Networks. Each residual block consists of two layers, as conventional.

All networks were trained using Adam (Kingma & Ba (2014)) with Nesterov momentum (Dozat) for a total of 100 epochs using mini-batches of size 128. No learning rate decay was used: we kept the learning rate and momentum fixed to 0.002 and 0.9 during the whole training.

For preprocessing, we divided each pixel value by 255, normalizing their values to $[0, 1]$.

The training curves for plain networks, Gated PlainNets, ResNets and Gated ResNets with varying depth are shown in Figure 4. The distance between the curves increase with the depth, showing that the augmentation helps the training of deeper models.

Table 1 shows the test error for each depth and architecture. Augmented models perform better in all settings when compared to the original ones, and the performance boost is more noticeable with increased depths. Interestingly, Gated PlainNets performed better than ResNets, suggesting that the reason for Highway Neural Networks to underperform ResNets might be due to an overly complex gating mechanism.

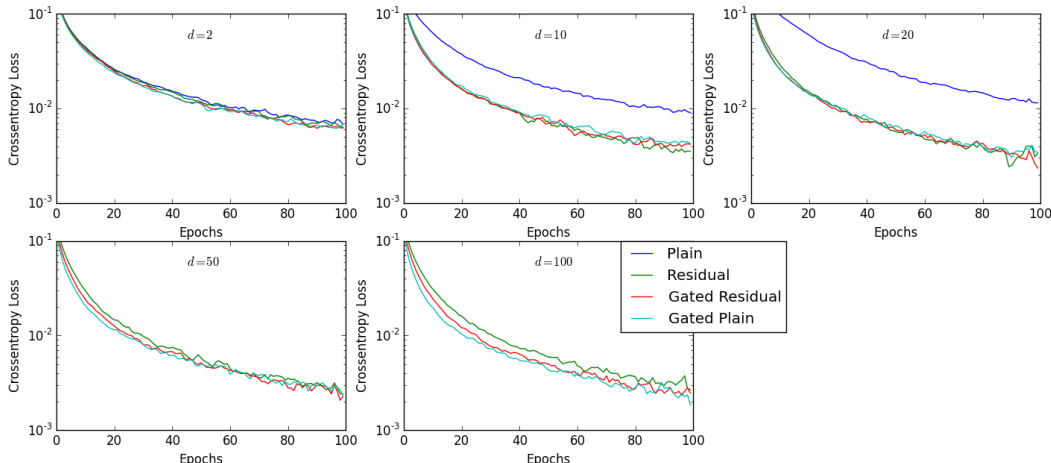

Figure 4: Train loss for plain and residual networks, along with their augmented counterparts, with $d = \{2, 10, 20, 50, 100\}$. As the models get deeper, the error reduction due to the augmentation increases.

| Depth = $d + 2$ | Plain | ResNet | Gated PlainNet | Gated ResNet |
|---|---|---|---|---|
| $d = 2$ | 2.29 | 2.20 | 2.04 | 2.17 |
| $d = 10$ | 2.22 | 1.64 | 1.78 | 1.60 |
| $d = 20$ | 2.21 | 1.61 | 1.59 | 1.57 |
| $d = 50$ | 60.37 | 1.62 | 1.36 | 1.48 |
| $d = 100$ | 90.20 | 1.50 | 1.29 | 1.26 |

Table 1: Test error (%) on the MNIST dataset for fully-connected networks. Augmented models outperform their original counterparts in all experiments. Non-augmented plain networks perform worse and fail to converge for $d = 50$ and $d = 100$.

| Depth = $d + 2$ | Gated PlainNet | Gated ResNet |
|---|---|---|
| $d = 2$ | 10.57 | 5.58 |
| $d = 10$ | 1.19 | 2.54 |
| $d = 20$ | 0.64 | 1.73 |
| $d = 50$ | 0.46 | 1.04 |
| $d = 100$ | 0.41 | 0.67 |

Table 2: Mean $k$ for increasingly deep Gated PlainNets and Gated ResNets.

As observed in Table 2, the mean values of $k$ decrease as the model gets deeper, showing that shortcut connections have less impact on shallow networks. This agrees with empirical results that ResNets perform better than classical plain networks as the depth increases. Note that the significant difference between mean values for $k$ in Gated PlainNets and Gated ResNets has an intuitive explanation: in order to suppress the residual signal against the shortcut connection, Gated PlainNets require that $k < 0.5$ (otherwise the residual signal will be enhanced). Conversely, Gated ResNets suppress the residual signal when $k < 1.0$, and enhance it otherwise.

We also analyzed how layer removal affects ResNets and Gated ResNets. We compared how the deepest networks ($d = 100$) behave as residual blocks composed of 2 layers are completely removed from the models. The final values for each $k$ parameter, according to its corresponding residual block, is shown in Figure 5. We can observe that layers close to the middle of the network have a smaller $k$ than these in the beginning or the end. Therefore, the middle layers have less importance by due to being closer to identity mappings.

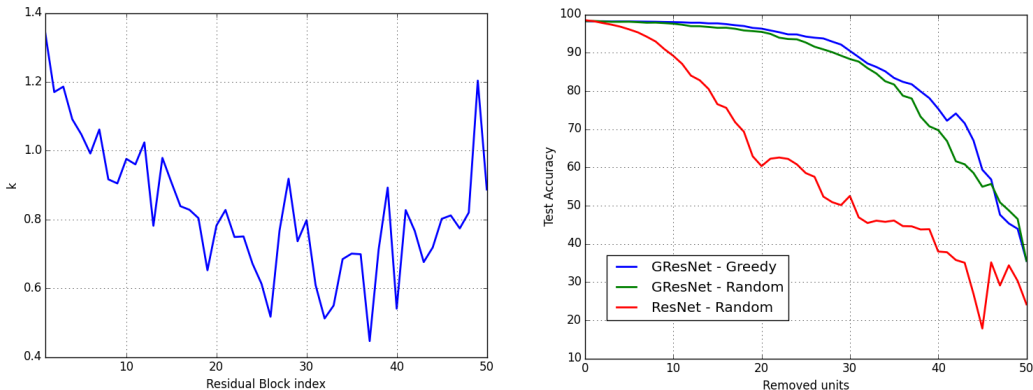

Figure 5: *Left:* Values for $k$ according to ascending order of residual blocks. The first block, consisted of the first two layers of the network, has index 1, while the last block – right before the softmax layer – has index 50. *Right:* Test accuracy (%) according to the number of removed layers. Gated Residual Networks are more robust to layer removal, and maintain decent results even after half of the layers have been removed.

Results are shown in Figure 5. For Gated Residual Networks, we prune pairs of layers following two strategies. One consists of pruning layers in a greedy fashion, where blocks with the smallest $k$ are removed first. In the other we remove blocks randomly. We present results using both strategies for Gated ResNets, and only random pruning for ResNets since they lack the $k$ parameter.

The greedy strategy is slightly better for Gated Residual Networks, showing that the $k$ parameter is indeed a good indicator of a layer's importance for the model, but that layers tend to assume the same level of significance. In a fair comparison, where both models are pruned randomly, Gated ResNets retain a satisfactory performance even after half of its layers have been removed, while ResNets suffer performance decrease after just a few layers.

Therefore augmented models are not only more robust to layer removal, but can have a fair share of their layers pruned and still perform well. Faster predictions can be generated by using a pruned version of an original model.

## 3.2 CIFAR

The CIFAR datasets (Krizhevsky (2009)) consists of $60,000$ color images with $32 \times 32$ pixels each. CIFAR-10 has a total of 10 classes, including pictures of cats, birds and airplanes. The CIFAR-100 dataset is composed of the same number of images, however with a total of 100 classes.

Residual Networks have surpassed state-of-the-art results on CIFAR. We test Gated ResNets, Wide Gated ResNets (Zagoruyko & Komodakis (2016)) and compare them with their original, non-augmented models.

For pre-activation ResNets, as described in He et al. (2016), we follow the original implementation details. We set an initial learning rate of 0.1, and decrease it by a factor of 10 after 50% and 75% epochs. SGD with Nesterov momentum of 0.9 are used for optimization, and the only pre-processing consists of mean subtraction. Weight decay of 0.0001 is used for regularization, and Batch Normalization's momentum is set to 0.9.

We follow the implementation from Zagoruyko & Komodakis (2016) for Wide ResNets. The learning rate is initialized as 0.1, and decreases by a factor of 5 after 30%, 60% and 80% epochs. Images are mean/std normalized, and a weight decay of 0.0005 is used for regularization. We also apply 0.3 dropout (Srivastava et al. (2014)) between convolutions, whenever specified. All other details are the same as for ResNets.

For both architectures we use moderate data augmentation: images are padded with 4 pixels, and we take random crops of size $32 \times 32$ during training. Additionally, each image is horizontally flipped with $50\%$ probability. We use batch size 128 for all experiments.

For all gated networks, we initialize $k$ with a constant value of 1. One crucial question is whether weight decay should be applied to the $k$ parameters. We call this "$k$ decay", and also compare Gated ResNets and Wide Gated ResNets when it is applied with the same magnitude of the weight decay: 0.0001 for Gated ResNet and 0.0005 for Wide Gated ResNet.

| Model | Original | Gated | Gated ($k$ decay) |
|---|---|---|---|
| Resnet 5 | 7.16 | **6.67** | 7.04 |
| Wide ResNet (4,10) + Dropout | 3.89 | **3.65** | 3.74 |

Table 3: Test error (%) on the CIFAR-10 dataset, for ResNets, Wide ResNets and their augmented counterparts. $k$ decay is when weight decay is also applied to the $k$ parameters in an augmented network. Results for the original models are as reported in He et al. (2015b) and Zagoruyko & Komodakis (2016).

Table 3 shows the test error for two architectures: a ResNet with $n = 5$, and a Wide ResNet with $n = 4$, $n = 10$. Augmenting each model adds 15 and 12 parameters, respectively. We observe that $k$ decay hurts performance in both cases, indicating that they should either remain unregularized or suffer a more subtle regularization compared to the weight parameters. Due to its direct connection to layer degeneration, regularizing $k$ results in enforcing identity mappings, which might harm the model.

Due to the indications that a regularization on the $k$ parameter results in a negative impact on the model's performance, we proceed to test other models – having different depths and widening factors – with the goal of evaluating the effectiveness of our proposed augmentation. Tables 4 and 5 show that augmented Wide ResNets outperform the original models without changing any hyperparameter, both on CIFAR-10 and CIFAR-100.

| Model | Original | Gated |
|---|---|---|
| Wide ResNet (2,4) | 5.02 | **4.66** |
| Wide ResNet (4,10) | 4.00 | **3.82** |
| Wide ResNet (4,10) + Dropout | 3.89 | **3.65** |
| Wide ResNet (8,1) | 6.43 | **6.10** |
| Wide ResNet (6,10) + Dropout | 3.80 | **3.63** |

Table 4: Test error (%) on the CIFAR-10 dataset, for Wide ResNets and their augmented counterparts. Results for non-gated Wide ResNets are from Zagoruyko & Komodakis (2016).

| Model | Original | Gated |
|---|---|---|
| Wide ResNet (2,4) | 24.03 | **23.29** |
| Wide ResNet (4,10) | 19.25 | **18.89** |
| Wide ResNet (4,10) + Dropout | 18.85 | **18.27** |
| Wide ResNet (8,1) | 29.89 | **28.20** |

Table 5: Test error (%) on the CIFAR-100 dataset, for Wide ResNets and their augmented counterparts. Results for non-gated Wide ResNets are from Zagoruyko & Komodakis (2016).

As in the previous experiment, in Figure 6 we present the final $k$ values for each block, in this case of the Wide ResNet (4,10) on CIFAR-10. We can observe that the $k$ values follow an intriguing pattern: the lowest values are for the blocks of index 1, 5 and 9, which are exactly the ones that increase the feature map dimension. This indicates that, in such residual blocks, the convolution performed in the shortcut connection to increase dimension is more important than the residual block itself. Additionally, the peak value for the last residual block suggests that its shortcut connection is of little importance, and could as well be fully removed without greatly impacting the model.

Figure 7 shows the loss curves for Gated Wide ResNet (4,10) + Dropout, both on CIFAR-10 and CIFAR-100. The optimization behaves similarly to the original model, suggesting that the gates do

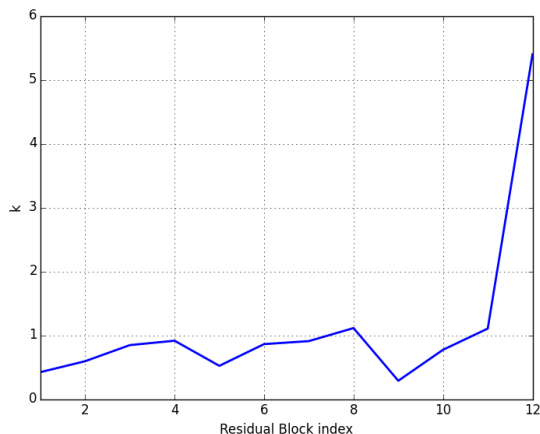

Figure 6: Values for $k$ according to ascending order of residual blocks. The first block, consisted of the first two layers of the network, has index 1, while the last block – right before the softmax layer – has index 12.

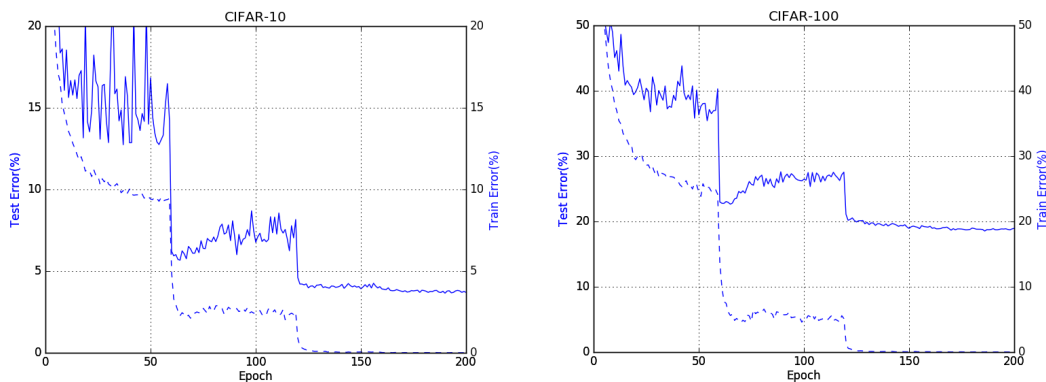

Figure 7: Training and test curves for the Wide ResNet (4,10) with 0.3 dropout, showing error (%) on training and test sets. Dashed lines represent training error, whereas solid lines represent test error.

not have any side effects on the network. The performance gains presented on Table 4 point that, however predictable and extremely simple, our augmentation technique is powerful enough to aid on the optimization of state-of-the-art models.

Results of different models on the CIFAR datasets are shown in Table 6. The training and test errors are presented in Figure 7. To the authors' knowledge, those are the best results on CIFAR-10 and CIFAR-100 with moderate data augmentation – only random flips and translations.

## 3.3 INTERPRETATION

Greff et al. (2016) showed how Residual and Highway layers can be interpreted as performing iterative refinements on learned representations. In this view, there is a connection on a layer's learned parameters and the level of refinement applied on its input: for Highway Neural Networks, $T(x)$ having components close to 1 results in a layer that generates completely new representations. As seen before, components close to 0 result in an identity mapping, meaning that the representations are not refined at all.

| Method | Params | C10+ | C100+ |
|---|---|---|---|
| Network in Network (Lin et al. (2013)) | - | 8.81 | - |
| FitNet (Romero et al. (2014)) | - | 8.39 | 35.04 |
| Highway Neural Network (Srivastava et al. (2015)) | 2.3M | 7.76 | 32.39 |
| All-CNN (Springenberg et al. (2014)) | - | 7.25 | 33.71 |
| ResNet-110 (He et al. (2015b)) | 1.7M | 6.61 | - |
| ResNet in ResNet (Targ et al. (2016)) | 1.7M | 5.01 | 22.90 |
| Stochastic Depth (Huang et al. (2016a)) | 10.2M | 4.91 | - |
| ResNet-1001 (He et al. (2016)) | 10.2M | 4.62 | 22.71 |
| FractalNet (Larsson et al. (2016)) | 38.6M | 4.60 | 23.73 |
| Wide ResNet (4,10) (Zagoruyko & Komodakis (2016)) | 36.5M | 3.89 | 18.85 |
| DenseNet (Huang et al. (2016b)) | 27.2M | 3.74 | 19.25 |
| Wide GatedResNet (4,10) + Dropout | 36.5M | **3.65** | **18.27** |

Table 6: Test error (%) on the CIFAR-10 and CIFAR-100 dataset. All results are with standard data augmentation (crops and flips).

However, the dependency of $T(x)$ on the incoming data makes it difficult to analyze the level of refinement performed by a layer given its parameters. This is more clearly observed once we consider how each component of $T(x)$ is a function not only on the parameter set $W_T$, but also on $x$.

In particular, given the mapping performed by a layer, we can estimate how much more abstract its representations are compared to the inputs. For our technique, this estimation can be done by observing the $k$ parameter of the corresponding layer: in Gated PlainNets, $k = 0$ corresponds to an identity mapping, and therefore there is no modification on the learned representations. For $k = 1$, the shortcut connection is ignored and therefore a jump in the representation's complexity is observed.

For Gated ResNets, the shortcut connection is never completely ignored in the generation of output. However, we can see that as $k$ grows to infinity the shortcut connection's contribution goes to zero, and the learned representation becomes more abstract compared to the layer's inputs.

Table 6 shows how the layers that change the data dimensionality learn more abstract representations compared to dimensionality-preserving layers, which agrees with Greff et al. (2016). The last layer's $k$ value, which is the biggest among the whole model, indicates a severe jump in the abstraction of its representation, and is intuitive once we see the model as being composed of two main stages: a convolutional one and a fully-connected one, specific for classification.

Finally, Table 2 shows that the abstraction jumps decrease as the model grows deeper and the performance increases. This agrees with the idea that depth allows for more refined representations to be learned. We believe that an extensive analysis on the rate that these measures – depth, abstraction jumps and performance – interact with each other could bring further understanding on the practical benefits of depth in networks.

## 4 CONCLUSION

We have proposed a novel layer augmentation technique that facilitates the optimization of deep networks by making identity mappings easy to learn. Unlike previous models, layers augmented by our technique require optimizing only one parameter to degenerate into identity, and by designing our method such that randomly initialized parameter sets are always close to identity mappings, our design offers less optimization issues caused by depth.

Our experiments showed that augmenting plain and residual layers improves performance and facilitates learning in settings with increased depth. In the MNIST dataset, augmented plain networks outperformed ResNets, suggesting that models with gated shortcut connections – such as Highway Neural Networks – could be further improved by redesigning the gates.

We have shown that applying our technique to ResNets yield a model that can regulate the residuals. This model performed better in all our experiments with negligible extra training time and

parameters. Lastly, we have shown how it can be used for layer pruning, effectively removing large numbers of parameters from a network without necessarily harming its performance.

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
