# Peer review of "Learning Identity Mappings with Residual Gates"

_ICLR 2017 — rejected_

[Reviewer Comment · AnonReviewer2 · 03 Dec 2016]
**metric and experiments**

- "assume that the squared parameter-wise distance between .. surrogate to the paths length". Can you elaborate more on this? Why should this be the case? 

- Why do you substitute mean and variance with initialization mean variance in derivation in Eq 15 and 16?

-  Regarding "A comparison of the Total Squared Distance to Identity…": where is the comparison?

- Why there is no Highway Neural Networks result in Table 5? 

- In Table 5, why does  He et al. (2015b)  give 6.61 which does not match with 6.43 in that paper? 

- Regarding "Our results don’t surpass the state-of-the-art, which was expected considering the hardware limitations. However, taking into account the improvement observed when augmenting a smaller Wide ResNet, we believe that the technique proposed can be used to surpass the state-of-the-art", you may use that result for sanity check during the empirical work but how can you claim that you will get any improvement over baselines in final comparison? 

- Baselines have Imagenet results. In order to obtain satisfactory comparison, you would need that too. Are there any results on Imagenet comparison?

[Public Comment · Pedro Savarese · 16 Dec 2016]
**Revision**

In the recent revisions of our work, we have performed the following main changes:

- Run complex models on CIFAR-10 and on CIFAR-100, achieving 3.65% and 18.27% test error, respectively. We have then removed our belief that the technique could achieve SOTA results, since now we have empirical indications of it.
- Added evaluation of the final k parameters for fully-connected and convolutional networks, showing interesting patterns for both architectures.
- Added layer pruning experiment to Wide GResNet.
- Made the intuition behind the distance to identity clearer.

We thank all the received feedback.

[Official Review · AnonReviewer1 · rating 6 · confidence 4 · 16 Dec 2016]
**Interesting approach for optimizing network architecture**

The paper presents a layer architecture where a single parameter is used to  gate the output response of layer to amplify or suppress it. It is shown that such an architecture can ease optimization of a deep network as it is easy to learn identity mappings in layers helping in better gradient propagation to lower layers (better supervision). 

Using an introduced SDI metric it shown that gated residual networks can most easily learn identity mappings compared to other architectures. 

Although good theoretical reasoning is presented the observed experimental evidence of learned k values does not seem to strongly support the theory given that learned  k values are mostly very small and not varying much across layers. Also, experimental validation of the approach is not quite strong in terms of reported performances and number of large scale experiments.

[Official Review · AnonReviewer3 · rating 5 · confidence 5 · 17 Dec 2016]
**A simple gating mechanism**

This paper proposes to learn a single scalar gating parameter instead of a full gating tensor in highway networks. The claim is that such gating is easier to learn and allows a network to flexibly utilize computation.

The basic idea of the paper is simple and is clearly presented. It is a natural simplification of highway networks to allow easily "shutting off" layers while keeping number of additional parameters low. However, in this regard the paper leaves out a few key points. Firstly, it does not mention that the gates in highway networks are data-dependent which is potentially more powerful than learning a fixed gate for all units and independent of data. Secondly, it does not do a fair comparison with highway networks to show that this simpler formulation is indeed easier to learn.

Did the authors try their original design of u = g(k)f(x) + (1 - g(k))x where f(x) is a plain layer instead of a residual layer? Based on the arguments made in the paper, this should work fine. Why wasn't it tested? If it doesn't work, are the arguments incorrect or incomplete?

For the MNIST experiments, since the hyperparameters are fixed, the plots are misleading if any dependence on hyperparameters exists for the different models. This experiment appears to be based on Srivastava et al (2015). If it is indeed designed to test optimization at aggressive depths, then apart from doing a hyperparameter search, the authors should not use regularization such as dropout or batch norm, which do not appear in the theoretical arguments for the architecture.

For CIFAR experiments, the obtained improvements compared to the baseline (wide resnets) are very small and therefore it is important to report the standard deviations (or all results) in both cases. It's not clear that the differences are significant.

Some questions regarding g(): Was g() always ReLU? Doesn't this have potential problems with g(k) becoming 0 and never recovering? Does this also mean that for the wide resnet in Fig 7, most residual blocks are zeroed out since k < 0?

[Official Review · AnonReviewer2 · rating 5 · confidence 5 · 20 Dec 2016 (modified: 23 Jan 2017)]
**claims not convincing**

This paper proposes a network called Gated Residual Networks layer design that adds gating to shortcut connections with a scalar to regulate the gate. The authors claim that this approach will improve the training Residual Networks.

It seems the authors could get competitive performance on CIFAR-10 to state of art models with only Wide Res Nets. Wide Gated ResNet requires much more parameters than DenseNet (and other Res Net variants) for obtaining a little improvement over Dense Net.  More importantly, the authors state that they obtained the best results on CIFAR-10 and CIFAR-100 but the updated version of DenseNet (Huang et al. (2016b)) has new results for a version called DenseNet-BC which outperforms all of the results that authors reported (3.46 for CIFAR-10 and 17.18 for CIFAR-100 with 25.6M parameters, DenseNet-BC still outperforms with 15.3M parameters which is much less that 36.5M). The Res Net variants papers with state of art results report result for Image Net. Therefore the empirical results need also the Image Net to demonstrate that improvement claimed is achieved.

The proposed trick adopts Highway Neural Networks and Residual Networks with an intuitive motivation. It is not sufficiently novel and the empirical results do not prove sufficient effectiveness of this incremental approach.

[Public Comment · Pedro Savarese · 24 Jan 2017]
**Revision**

In the last revision of our work, we have performed the following main changes:

- Added results with u = g(k)f(x) + (1 - g(k))x (Gated Plain Networks), along with some intuitions of why it outperformed non-augmented Residual Networks. This result suggests that an extensive study on different gating mechanisms for Highway Neural Networks can be extremely fruitful, once the original design is equivalent to a Highway Net with scalar gates. This also goes against the suggestions in the literature not to add gates to shortcut connections in order to keep an uncorrupted gradient flow through the network.

- Added Gated Plain Networks to the table with mean k values, along with an explanation for the significant difference when compared to mean k's of Gated ResNets.

- Added 'Understanding deep learning requires rethinking generalization' to bibliography.

- Rephrased a few parts when augmented models are compared to Highway Nets, showing more clearly the differences between the two designs. Also added a brief discussion regarding the impact for Highway Nets of the new results (Gated Plain Nets).


We are currently gathering results to introduce the following changes to the next revision:

- Add results for more aggressive depths (200+ layers), in order to better compare different models.

- Add results for Gated Plain Net on CIFAR.

We thank all the received feedback.

[Public Comment · Pedro Savarese · 30 Jan 2017]
**Revision**

In the last revision of our work, we have performed the following main changes:

- Added 6 extra experiments comparing Wide ResNet and their augmented counterparts (Table 4). With this we hope to provide stronger indications of the generality and advantages of our technique.


We are currently running the same 6 models on CIFAR-100 to add them to the table. Are have also run Gated PlainNets (u = g(k)f(x) + (1 - g(k))x) on CIFAR-10, as suggested by reviewer 3, and will add results after running on CIFAR-100 as well.


We thank all the received feedback.

[Public Comment · Pedro Savarese · 03 Feb 2017]
**Revision**

We have also added a section (3.3) that connects our technique to the Unrolled Iterative Estimation interpretation of Highway and ResNets. We further elaborate on how the learned k values are indicators of the abstraction jumps in representations defined in Greff et al, and analyze the k values for both experiments (MNIST and CIFAR) under this perspective.

[Public Comment · Pedro Savarese · 03 Feb 2017]
**Revision**

In the last revision of our work, we have performed the following main changes:

- Added 4 extra experiments comparing Wide ResNet and their augmented counterparts (Table 5) on CIFAR-100. With this we hope to provide stronger indications of the generality and advantages of our technique.

We are currently running Gated PlainNets on CIFAR-100.

We thank all the received feedback.

[Final Decision · Program Chairs · 06 Feb 2017]
**ICLR committee final decision**

Although this was a borderline paper, the reviewers ultimately concluded that, given how easy it would be for a practitioner to independently devise the methodological trick of the paper, the paper did not demonstrate that the idea was sufficiently useful to merit acceptance.